# ACCURATE AND EFFICIENT SINGULAR VALUE DECOMPOSITION FOR LLMS VIA DECAY-AWARE RANK ALLOCATION AND FEATURE-PRESERVED WEIGHT UPDATE

## ABSTRACT

Singular Value Decomposition (SVD) provides a hardware-agnostic and effective paradigm for compressing and accelerating Large Language Models (LLMs) by decomposing and truncating weight matrices, followed by weight updates to restore accuracy. However, SVD-based compression faces two major challenges: **(1) Rank Selection Problem:** Optimizing truncation and update ranks constitutes a high-dimensional combinatorial problem. Existing solutions rely on computationally expensive search, leading to both suboptimal performance and diminished efficiency. **(2) Limited Accuracy Restoration:** The sequential weight update strategy employed by state-of-the-art approaches (e.g., SVD-LLM) results in Hessian anisotropic, which hampers accuracy recovery and slows convergence. To overcome these, we introduce DF-SVD, which integrates: **(1) Decay-Aware Rank Allocation:** We derive and validate a correlation between decay characteristics of each weight's singular value spectrum and its importance. This enables dynamic, layer- and weight-specific rank allocation, ensuring high fidelity without costly search. **(2) Feature-Preserved Weight Update:** We introduce a theoretically grounded update strategy that fixes the truncated weight matrix $V^\top S^{-1}$ along with the principal components of $U\Sigma$, while updating only the minor components. This design ensures Hessian isotropic, achieving superior accuracy restoration and faster convergence. DF-SVD not only significantly outperforms baselines in accuracy, but also completing compression in just 30 minutes, achieving speedups of $7\times$, $11\times$ and $16\times$ compared to SVD-LLM, ASVD and Dobi-SVD respectively. DF-SVD directly correlates the singular spectrum with training-free rank selection and boosts Hessian isotropy, paving the way for a new paradigm in accurate and efficient SVD-based LLM compression.

## 1 INTRODUCTION

Singular Value Decomposition (SVD) Golub et al. (1987) has emerged as a promising post-training compression technique for Large Language Models (LLMs), offering distinct advantages over conventional approaches Zhu et al. (2024); Gao et al. (2024). Unlike quantization Dettmers et al. (2023); Shao et al. (2023) and unstructured pruning Sun et al. (2024) that require specialized hardware, or structured pruning An et al. (2024); Ma et al. (2024) and distillation Hsieh et al. (2023); Snell et al. (2024) that demand substantial training resources Yuan et al. (2024b), SVD provides a hardware-agnostic solution while maintaining model performance. Moreover, it can significantly enhance inference speed, making it particularly attractive for deployment scenarios NVIDIA (2007).

The prevailing SVD-based LLM compression involves decomposing weight matrices followed by truncation, with subsequent weight updates to recover model accuracy Hsu et al. (2022); Yuan et al. (2024a); Wang et al. (2024). While effective, this methodology faces two critical limitations. First, the rank selection problem presents a significant bottleneck. Determining the optimal truncation and update ranks for each weight matrix—each with varying importance across different layers and modules—constitutes a high-dimensional combinatorial optimization problem Gao et al. (2024); Lagunas et al. (2021a); Sharma et al. (2024); Zhang et al. (2023). Existing solutions, such as reinforcement learning Schulman et al. (2017) or differentiable search frameworks Qinsi et al. (2025);

Gao et al. (2024); Liu et al. (2024), attempt to address this. This not only leads to suboptimal compression performance but also severely undermines the efficiency Li et al. (2025).

Second, current state-of-the-art SVD-based LLMs compression method SVD-LLM Wang et al. (2024) employs sequential low-rank optimization of truncated weights to restore model accuracy, shown in Figure 1 (top), where $W_u = U\sqrt{\Sigma}$ and $W_v = \sqrt{\Sigma}V^\top S^{-1}$. They first freeze matrix $W_u$ and fine-tune $W_v$, then updating the matrix $W_v$ while freezing the updated matrix $W_u$. However, our theoretical analysis reveals a fundamental limitation: the Hessian condition number of this updating strategy is inherently constrained by the anisotropic distribution of singular values. Moreover, this method suffer from destructive parametric perturbation, as it relies on conventional low-rank decomposition techniques Hu et al. (2021), where low-rank matrices $A$ and $B$ are initialized with Gaussian noise and zeros. This initialization risks degrading pre-trained features in compressed model, which can significantly impair convergence Meng et al. (2024); Wang et al. (2025); Liu et al. (2025).

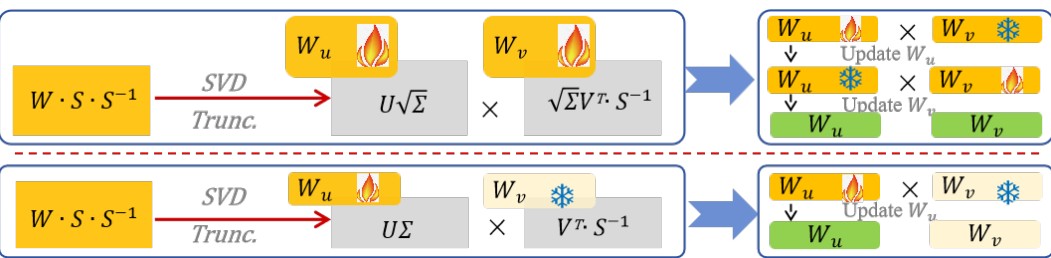

Figure 1: Comparison between weight updating between SVD-LLM (top) and DF-SVD (bottom). $S$ refers the whitening matrix obtained by Cholesky decomposition through input activation.

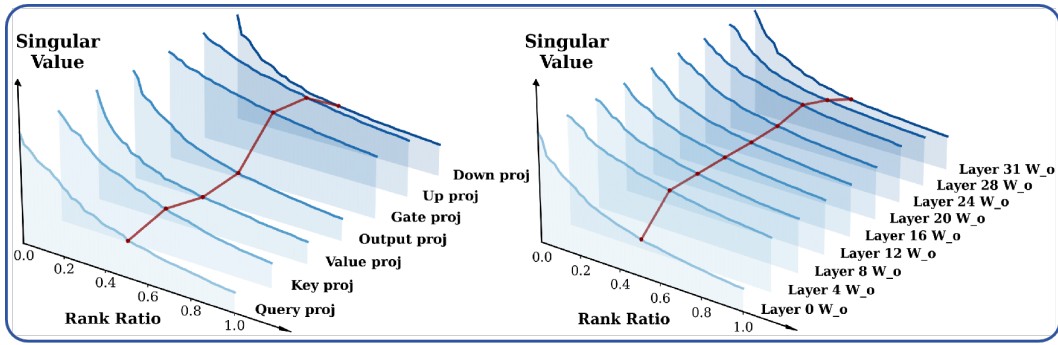

Figure 2: Singular Value of different weights and layers. Left: Spectrum of different weight matrices within the same layer. Right: Spectrum of the same weight matrices of different layers.

Regarding the first challenge, we hypothesize a latent correlation between weight matrix importance and decay characteristics of singular value: matrices encoding fewer salient features exhibit singular values concentrated in leading components, while those representing richer feature spaces show more uniform spectral distributions. To validate this, we systematically compare singular value spectra across weight matrices under two scenarios. Figure 2 (left) compares distributions across modules within the same layer, revealing feed-forward layers exhibit significantly slower decay than attention layers. Similarly, Figure 2 (right) tracks spectral evolution across network depth, showing deeper layers maintain singular values more effectively, with markedly slower decay. These results aligns closely with AdaLoRA Zhang et al. (2023). In this work, we assume that the singular values of LLM weight matrices approximately follow an exponential decay, which forms the basis of our decay-aware rank allocation scheme. Appendix A.1 provides empirical evidence supporting this assumption through comparisons with logarithmic and power-law alternatives, while Appendix A.2 offers theoretical justification grounded in operator theory and approximation theory. These observations are consistent with prior findings on spectral decay behaviors Thamm et al. (2022);

Plerou et al. (2002).Based on these insights, we propose that dynamically allocating the truncation and updating ranks based on the decay characteristics of singular value spectrum may further optimize model performance while eliminating complex search and training procedures. For the second challenge, our theoretical analysis indicates that a significant portion of the truncated matrices do not require updates, yet proper initialization is essential. Inappropriate matrix updates or initialization may result in insufficient recovery of model performance and long recovery times.

Therefore, based on the foundations outlined above, we propose DF-SVD to address these challenges, which integrates the following key components: **(1) Decay-Aware Rank Allocation:** We dynamically determines both the truncation position and weight updating ranks for individual weight matrices by analyzing the decay characteristics of their singular value spectrum. This intelligent rank allocation is theoretically supported, significantly enhancing model performance while introducing minimal computational overhead. **(2) Feature-Preserved Weight Update:** We decompose truncated weight matrices into two components: $W_u = U\Sigma$ and $W_v = V^\top S^{-1}$. During the weight updating phase, we employ a selective updating scheme where $W_v = V^\top S^{-1}$ remains frozen while exclusively updating $W_u = U\Sigma$ to restore accuracy (see Figure 1, bottom). Furthermore, we preserve the pre-trained features by fixing the principal components of $W_u = U\Sigma$ while only updating its minor components. Theoretical analysis shows this method produces an isotropic Hessian matrix, guaranteeing both optimal accuracy restoration and rapid convergence.

Extensive experiments demonstrate DF-SVD's dual breakthrough: it significantly improves model accuracy over existing SVD-based methods while completing the entire compression process in just 30 minutes - achieving $7\times$, $11\times$, and $16\times$ speedups over SVD-LLM, ASVD, and Dobi-SVD respectively. This unique combination of superior performance and unprecedented efficiency establishes DF-SVD as the leading post-training compression method for practical LLMs deployment.

## 2 RELATED WORKS

### 2.1 LARGE LANGUAGE MODEL COMPRESSION

LLMs compression methods including distillation, pruning, and quantization. Distillation includes DISTILLSPEC Zhou et al. (2024), LLMA Yang et al. (2023), training a smaller to mimic the behavior of LLMs but need substantial compute for retraining. Pruning includes LLM-Pruner Ma et al. (2024), SliceGPT Ashkboos et al. (2024), and BlockPrune Lagunas et al. (2021b), removing channels or components but often cause significant accuracy loss. Quantization includes SmoothQuant Xiao et al. (2023), LLM-int8 Dettmers et al. (2024), OS+ Wei et al. (2023), employing reduced numerical precision for weight parameters, minimizing the model's memory usage. However, it necessitates specialized kernels to achieve efficient inference. In contrast, SVD effectively compresses and accelerates LLMs without retraining or specialized hardware.

### 2.2 SVD-BASED LLMS COMPRESSION METHODS

There are numerous SVD-based LLMs compression methods. FWSVD Hsu et al. (2022) leverages Fisher information to evaluate the significance of each parameter. ARS Gao et al. (2024) proposes a binary masking mechanism for adaptive SVD rank selection. Yu & Wu (2023) observes weights are full-rank, making decomposition prone to accuracy loss, whereas features tend to be low-rank; thus, ASVD Yuan et al. (2024a) improves accuracy by decomposing features instead. Additionally, SVD-LLM Wang et al. (2024) utilizes Cholesky decomposition to establish a relationship between singular values and compression loss, further improving the accuracy. However, they suffer from two main drawbacks: (1) They apply uniform truncation and updating ranks across all layers and weight matrices, while overlooking their varying significance, leading to suboptimal performance; and (2) The entire compression process is inefficient, leading to low accuracy and prolonged runtime, significantly limiting practical adoption compared to traditional post-training compression methods.

### 2.3 ADAPTIVE RANK ALLOCATION

Recent studies have investigated methods for determining the optimal rank configuration of individual weight matrices and layers in neural networks. Reinforcement learning approaches Schulman et al. (2017) and evolutionary algorithms Real et al. (2019) offer potential solutions, however, they

introduce substantial computational overhead, making them impractical for large-scale models. Alternative differentiable frameworks Gao et al. (2024); Liu et al. (2024) have been proposed to mitigate this issue. For example, ARS Gao et al. (2024) proposes a novel binary masking mechanism for optimizing the number of ranks in a differentiable framework. AdaSVD Li et al. (2025) assigns layer-specific compression ratios according to their relative importance, however, weight-specific significance should also be considered. Dobi-SVD Qinsi et al. (2025) proposes a differentiable truncation mechanism, along with gradient-robust backpropagation to adaptively find the optimal truncation positions. However, their extensive training requirements not only hinder performance but also undermine the core efficiency advantages that post-training techniques aim to deliver.

## 3    DF-SVD

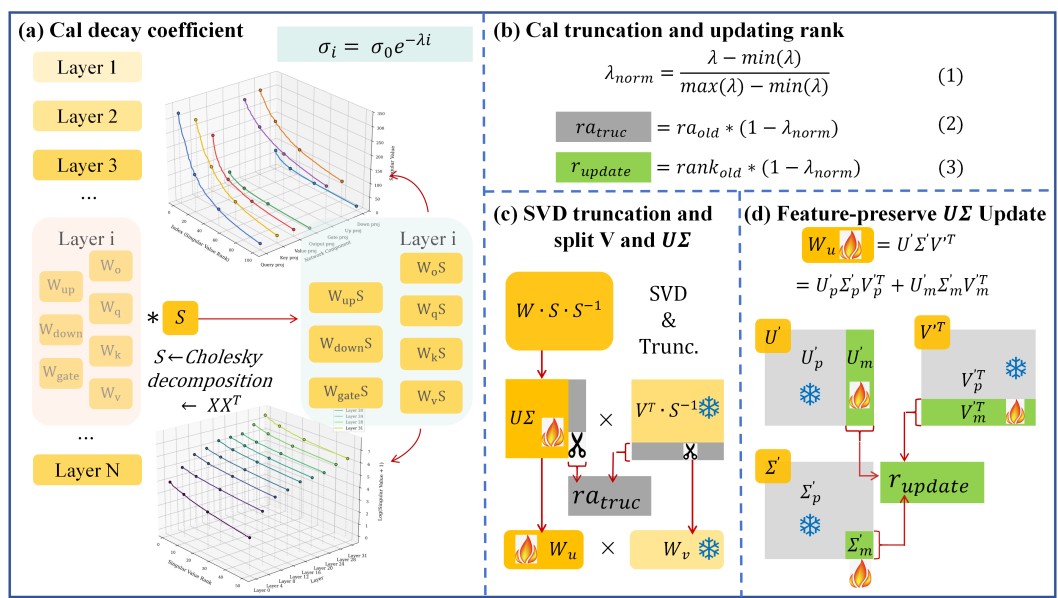

Figure 3: Overview of DF-SVD. Pipeline is shown in Algorithm 1

### 3.1    DECAY-AWARE RANK ALLOCATION

To quantitatively characterize the decay characteristics across different weight matrices, we employ an exponential decay model as a powerful approximator. This approach provides a smooth, differentiable proxy for the underlying spectral trends, enabling the extraction of a single, comparable decay coefficient for each matrix, which is fundamental to our dynamic rank allocation strategy.

First, building upon SVD-LLM, we initiate with Cholesky decomposition of the feature covariance matrix to derive an invertible transformation matrix S that encapsulates contextual information:

$$S = \text{Cholesky}(XX^\top) \tag{1}$$

where $X$ represents the input activation of the weight matrix $W$. Then we transform the weight matrix into a decomposition-friendly matrix $WS$. Subsequently, SVD is performed on $WS$ to obtain the decomposed matrices. Then, we obtain the decomposed weight matrix:

$$W = WSS^{-1} = U \times \Sigma \times V \times S^{-1} \tag{2}$$

Formally, we model singular value spectrum of $L$ weight matrices using exponential decay pattern:

$$\sigma_i = \sigma_0 e^{-\lambda i} \tag{3}$$

where $\sigma_i$ denotes the $i$-th singular value, $\sigma_0$ represents the initial singular value amplitude, and $\lambda$ is the module-specific decay rate coefficient. Then, to enable cross-module comparison, the decay

coefficients $\lambda$ of different weights are normalized to a standardized range [0,1]:

$$\lambda_{\text{norm}} = \frac{\lambda - \min(\lambda)}{\max(\lambda) - \min(\lambda)} \tag{4}$$

where $\min(\lambda)$ represents the minimum decay coefficient across all modules, $\max(\lambda)$ represents maximum decay coefficient across all modules, and $\lambda_{\text{norm}}$ represents the normalized decay coefficient of individual weight matrices, values near 1 indicate rapid decay and 0 signify slow decay.

Then, the normalized decay coefficients $\lambda_{\text{norm}}$ are used to dynamically adjust the truncation position and updating rank in the truncation and parameter updating process simultaneously. Among them, in the truncation process, decay coefficient preserves more data from slowly decaying modules while aggressively compresses rapidly decaying ones. The adjusted truncation position is computed as:

$$ra_{\text{trunc}} = ra_{\text{old}} * (1 - \lambda_{\text{norm}}) \tag{5}$$

where $ra_{\text{old}}$ is the original truncation position. The truncation position $ra_{\text{old}}$ is determined by the global compression ratio. For a weight matrix $W$, the $ra_{\text{old}}$ is calculated as:

$$ra_{\text{old}} = \left\lfloor \frac{m \times n \times r}{m + n} \right\rfloor \tag{6}$$

where $m$ is the number of rows in matrix $W$, $n$ is the number of columns in matrix $W$, and $r$ is the compression ratio. $\lfloor \cdots \rfloor$ is the floor function, denoting the largest integer less than or equal to the argument. $ra_{\text{trunc}}$ scales the reverse ratio inversely with decay speed. Slow-decaying modules indicates a larger fraction is retained, and fast-decay modules indicating heavily compressed. Additionally, the rank in weight updating process is modified as:

$$r_{\text{update}} = \text{rank}_{\text{old}} * (1 - \lambda_{\text{norm}}) \tag{7}$$

where $\text{rank}_{\text{old}}$ is a fixed, low-rank dimension that is typically set to a user-specified value, and $r_{\text{update}}$ are the original and modified rank respectively in weight updating process. The design of updating rank follows the similar principle as the truncation rank: matrices with rapidly decaying singular values contain fewer significant features and thus require fewer trainable parameters, while matrices with slowly decaying spectra preserve more essential features and consequently need higher-rank approximations. To rigorously justify our decay-aware strategy, we analyze the energy retention ratio at rank k, defined as:

$$E(k) = \frac{\sum_{i=1}^{k} \sigma_i^2}{\sum_{i=1}^{r} \sigma_i^2} \tag{8}$$

Under the exponential decay assumption $\sigma_i = \sigma_0 \cdot e^{-\lambda i}$, we approximate the discrete energy sum via continuous integration:

$$\sum_{i=1}^{k} \sigma_i^2 \approx \int_0^k \sigma_0^2 e^{-2\lambda x} \, dx = \frac{\sigma_0^2}{2\lambda} (1 - e^{-2\lambda k}) \tag{9}$$

The total energy is thus:

$$E_{\text{total}} = \frac{\sigma_0^2}{2\lambda} \tag{10}$$

yielding the normalized energy ratio:

$$E(k) = 1 - e^{-2\lambda k} \tag{11}$$

By introducing the normalized decay rate into our dynamic rank allocation strategy, we compare the energy ratio with fixed-rank approaches:

$$\frac{E(r_{\text{update}})}{E(r_{\text{fix}})} = \frac{1 - e^{-2\lambda r_{\text{base}}(1 - \lambda_{\text{norm}})}}{1 - e^{-2\lambda r_{\text{fix}}}} \tag{12}$$

Differentiating with respect to $\lambda$ proves that:

$$\frac{d}{db} \left( \frac{E_{\text{dynamic}}}{E_{\text{fix}}} \right) > 0 \quad \forall \lambda \in (\lambda_{\text{min}}, \lambda_{\text{max}}) \tag{13}$$

This result demonstrates that our decay-aware rank allocation strategy consistently outperforms fixed-rank approaches under conditions of exponential spectral decay. It establishes a direct and efficient link between an easily computable property—the singular spectrum's decay—and optimal rank selection, thereby bypassing the need for computationally expensive search or optimization procedures. The pipeline of the decay-aware rank allocation method is outlined in Algorithm 2.

## 3.2 Feature-Preserved Weight Update

SVD-LLM Wang et al. (2024) decomposes the original weight matrix into two low-ranking matrices $W_u = U\sqrt{\Sigma}$ and $W_v = \sqrt{\Sigma}V^\top S^{-1}$, then treats $W_u$ and $W_v$ as two linear layers and update them sequentially as:

$$Y = W'_u \times W'_v \times X \tag{14}$$

where $W'_u = W_u + B_u A_u$, $W'_v = W_v + B_v A_v$, and $A_u$, $B_u$, $A_v$, and $B_v$ are matrices used for LoRA fine-tuning. To prevent counteract between the two matrices, they first freeze matrix $W_u$ and fine-tune $W_v$ with LoRA, then updating the matrix $W_v$ while freezing the updated weight matrix $W_u$. To analyze the optimization characteristics, we define the reconstruction error loss function:

$$\mathcal{L} = \|W_u W_v X - WX\|_F^2 \tag{15}$$

where $X \in \mathbb{R}^{n \times m}$ is the input data matrix. The gradient of the loss function with respect to $W_u$ is:

$$\frac{\partial \mathcal{L}}{\partial W_u} = 2(W_u W_v X - WX)(W_v X)^\top \tag{16}$$

The corresponding Hessian matrix is:

$$H = \frac{\partial^2 \mathcal{L}}{\partial W_u^2} = 2(W_v X)(W_v X)^\top \tag{17}$$

Since $S^{-1}X$ is orthogonal (i.e., $(S^{-1}X)(S^{-1}X)^\top = I$), we have:

$$W_v X = \sqrt{\Sigma}V^\top S^{-1}X = \sqrt{\Sigma}Q \tag{18}$$

where $QQ^\top = I$. The Hessian matrix is:

$$H_{svd-llm} = 2(\sqrt{\Sigma}Q)(\sqrt{\Sigma}Q)^\top = 2\Sigma QQ^\top = 2\Sigma \tag{19}$$

Its eigenvalues are $2\sigma_i$, and the condition number is:

$$\kappa(H_{svd-llm}) = \frac{\sigma_{\max}}{\sigma_{\min}} \tag{20}$$

The analysis reveals that the Hessian condition number depends directly on the singular value distribution, potentially leading to slow convergence when $\sigma_{\max} \gg \sigma_{\min}$. Based on this, we propose Feature-Preserved Weight Updating methods, which define matrices $W_u = U\Sigma$ and $W_v = V^\top S^{-1}$, additionally, it fixed matrix $W_v = V^\top S^{-1}$ while only update matrix $W_u = U\Sigma$ in the updating process, under this condition, we can get:

$$W_v X = V^\top S^{-1}X = Q \tag{21}$$

The Hessian matrix at this point is:

$$H_{DF-SVD} = 2QQ^\top = 2I \tag{22}$$

and the condition number is:

$$\kappa(H_{DF-SVD}) = 1 \tag{23}$$

Therefore, we concentrates all singular values in $W_u$, resulting in an isotropic Hessian $2I$ with optimal convergence properties. Then, we provide a theoretical link to the end-task loss. Under standard assumptions in neural network analysis, small perturbations in weights bound deviations in task loss. The output perturbation of a single layer is:

$$\|W'_u W_v X - WX\|_F \leq \|W'_u - W_u\|_F \|W_v X\|_F \tag{24}$$

Since $\|W_v X\|_F = \|Q\|_F$ and $QQ^\top = I$, $\|Q\|_F = \sqrt{r}$ (for square orthogonal matrices, but generalizes). Minimizing $\mathcal{L}_{recon}$ thus bounds $\|W' - W\|_F \leq \epsilon$, where $W' = W'_u W_v$.

For the full network with $L$ layers, assume each layer's post-activation is $K$-Lipschitz. The output logit deviation at the final layer is bounded by $O(\epsilon K^{L-1})$ (via chain rule on forward passes). For cross-entropy loss $\mathcal{L}_{task} = -\sum y \log(\sigma(z))$ (where $z$ are logits, $\sigma$ is softmax), the loss is 1-Lipschitz in $z$ (since softmax gradients are bounded). Thus:

$$|\mathcal{L}_{task}(z') - \mathcal{L}_{task}(z)| \leq \|z' - z\| \leq O(\epsilon K^{L-1}) \tag{25}$$

where $z'$ are perturbed logits. This shows that minimizing reconstruction error (to achieve small $\epsilon$) bounds task loss deviation, implying no inherent negative impact—rather, it supports task recovery, as evidenced by our empirical results. Potential limitations arise if truncation discards task-critical minor features, but our decay-aware rank allocation mitigates this by preserving more ranks in slowly decaying modules, ensuring adaptability.

Additionally, the low-rank matrices $A_u$, $B_u$, $A_u$ and $B_u$ are initialized with Gaussian noise or zeros, risking degrading pre-trained features in compressed model, which can significantly impair convergence during weight update process Wang et al. (2025). To overcome this, we keeps the principal components of $U\Sigma$, while only minor components of $U\Sigma$ are initialized as low-rank matrices, to retrain pretrained knowledge. To conveniently separate principal and minor components of $U\Sigma$, we decompose $U\Sigma$ into three matrices $U'$, $\Sigma'$, $V'$ through SVD technique:

$$U\Sigma = U'\Sigma'V'^{\top} \tag{26}$$

We divide $U\Sigma$ into two components: the principal matrix $W_p$ corresponding to large singular values and the minor matrix $W_m$ Corresponding to small singular values:

$$U'\Sigma'V'^{\top} = \underbrace{U'_p\Sigma'_p V'^{\top}_p}_{\text{Fixed Principal Components}} + \underbrace{U'_m\Sigma'_m V'^{\top}_m}_{\text{Updated Minor Components}} = W_p + W_m \tag{27}$$

We argue that the principal matrix captures the essence of the pretrained knowledge, whereas the minor matrix is suboptimal with noisy or long-tail information, which is supported by previous works Ben Noach & Goldberg (2020); Sharma et al. (2024); Li et al. (2024). Therefore, we freeze the principal matrix $W_p$ of $U\Sigma$ and adapt the minor singular components $W_p$ of $U\Sigma$ during finetuning:

$$W_m = U_m\Sigma_m V_m^{\top} = (U_m\sqrt{\Sigma_m})(\sqrt{\Sigma_m}V_m^{\top}) = B_m A_m \tag{28}$$

Among them, the updating rank low-rank matrices of $A_m$, $B_m$ of individual weights is based on $r_{update}$ calculated based on the decay coefficient of singular values (Eq 4). By fixing principal components $U_p\Sigma_p$ and only updating minor components $U_m\Sigma_m$, we constrain:

$$\sigma_{\max}(H_{\mathrm{m}}) \leq \sigma_{\max}(H_{(\mathrm{p+m})}), \quad \kappa(H_{\mathrm{m}}) \leq \kappa(H_{(\mathrm{p+m})}) \tag{29}$$

where $\sigma_{\max}(H_{\mathrm{m}})$ and $\sigma_{\max}(H_{\mathrm{p+m}})$ denotes the largest singular value of the Hessian matrix of $H_m$ and $H_{p+m}$, while $\kappa(H_{\mathrm{m}})$ and $\kappa(H_{\mathrm{m+p}})$ denotes the condition number of the Hessian matrix $H_{\mathrm{m}}$ and $H_{\mathrm{p+m}}$ respectively. Therefore, Feature-Preserved Weight update fixes matrix $V$ and the principal components of $U\Sigma$, while only updates the minor components of $U\Sigma$, enabling preserved pretrained knowledge and efficient accuracy restoration. The pipeline is shown in Algorithm 3.

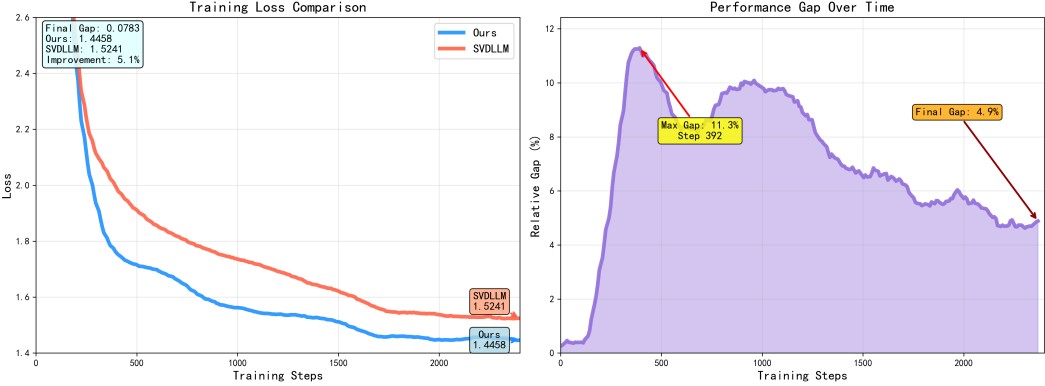

Figure 4: Convergence behavior of the weight update process for SVD-LLM and DF-SVD on LLaMA-7B. Left: training loss curves during the reconstruction fine-tuning stage. Right: relative performance gap between DF-SVD and SVD-LLM over training steps.

To empirically validate the proposed feature-preserved weight update, we compare DF-SVD with SVD-LLM on the reconstruction fine-tuning of LLaMA-7B. As shown in Figure 4, DF-SVD converges faster, maintains lower training loss, and establishes a clear performance advantage throughout optimization. These results on LLaMA-7B are consistent with our Hessian analysis, confirming that enforcing Hessian isotropy leads to more stable and effective weight updates.

# 4 EXPERIMENTS AND ANALYSIS

**Baselines.** We compare DF-SVD with current leading pruning methods (LLM-Pruner, Wanda) SVD-based LLMs compression methods FWSVD Hsu et al. (2022), ASVD Yuan et al. (2024a), SVD-LLM Wang et al. (2024), Dobi-SVD Qinsi et al. (2025), and AdaSVD Li et al. (2025).

**Model and Datasets.** We evaluate performance of DF-SVD and the baselines on three different LLM models, LLAMA 7B Touvron et al. (2023b), LLAMA2 7B Touvron et al. (2023a), LLAMA3 8B Grattafiori et al. (2024), and OPT 6.7B Zhang et al. (2022) and 8 datasets including two language modeling datasets (WikiText-2 Merity et al. (2016), C4 Raffel et al. (2020)) and six common sense reasoning datasets (OpenbookQA Mihaylov et al. (2018), ARC-e Clark et al. (2018), Hellaswag Zellers et al. (2019), WinoGrande Sakaguchi et al. (2019), PIQA Bisk et al. (2020) and MathQA Amini et al. (2019)) in zero-shot setting with the LM-Evaluation-Harness framework.

**Implementation Details.** We adopted the same method as SVD-LLM by randomly selecting 256 samples from Wikitext-2 for fair evaluation. Experiments were performed on NVIDIA A800 GPUs. Additionally, all reported time costs in these comparisons are end-to-end wall-clock times for the full compression pipeline, rather than partial or per-step timings.

## 4.1 COMPARISON WITH STATE-OF-THE-ART COMPRESSION METHODS

we have evaluated the performance of DF-SVD across six aspects: compression ratio performance, efficiency, SVD method comparison, cross-model generality, pruning baseline comparison, and quantization integration. Moreover, we have experimentally demonstrated that DF-SVD can significantly enhance inference efficiency on real hardware, as detailed in Appendix A.5.

**Performance under Different Compression Ratios.** We evaluate the performance of LLaMA-7B compressed at ratios ranging from 30% to 60% across eight datasets (Table 1). Among them, SVD-LLM applies decomposition alone, while SVD-LLM* includes both decomposition and update. Similarly, DF-SVD applies decomposition alone, while DF-SVD* includes both decomposition and update. DF-SVD almost outperforms all baseline methods at nearly every compression ratio.

Table 1: Performance of Llama-7B compressed by DF-SVD and baselines under 30% to 60% compression ratio in terms of perplexity and accuracy.

| RATIO | METHOD | WIKITEXT-2↓ | C4↓ | OPENB.↑ | ARC_E↑ | HELLAS.↑ | WINOG.↑ | PIQA↑ | MATHQA↑ | AVERAGE↑ |
|---|---|---|---|---|---|---|---|---|---|---|
| 0% | ORIGINAL | 5.68 | 7.34 | 0.28 | 0.67 | 0.56 | 0.67 | 0.78 | 0.27 | 0.53 |
| 30% | SVD | 13103 | 20871 | 0.13 | 0.26 | 0.26 | 0.51 | 0.54 | 0.22 | 0.32 |
| | FWSVD | 20127 | 7240 | 0.17 | 0.26 | 0.26 | 0.49 | 0.51 | 0.19 | 0.31 |
| | ASVD | 51 | 41 | 0.18 | 0.43 | 0.37 | 0.53 | 0.65 | 0.21 | 0.39 |
| | SVD-LLM | 9.56 | 25.11 | 0.20 | 0.48 | 0.37 | 0.59 | 0.65 | 0.22 | 0.42 |
| | SVD–LLM* | 8.13 | 12.95 | 0.26 | 0.68 | 0.47 | 0.64 | 0.71 | 0.24 | 0.50 |
| | **DF-SVD** | **9.05** | **21.13** | **0.23** | **0.60** | **0.39** | **0.64** | **0.67** | **0.23** | **0.46** |
| | **DF-SVD*** | **7.87** | **12.22** | **0.28** | **0.68** | **0.49** | **0.64** | **0.75** | **0.24** | **0.51** |
| 40% | SVD | 52489 | 47774 | 0.15 | 0.26 | 0.26 | 0.52 | 0.53 | 0.20 | 0.32 |
| | FWSVD | 18156 | 12847 | 0.16 | 0.26 | 0.26 | 0.51 | 0.53 | 0.21 | 0.32 |
| | ASVD | 1407 | 1109 | 0.13 | 0.28 | 0.26 | 0.48 | 0.55 | 0.19 | 0.31 |
| | SVD-LLM | 13.73 | 75.42 | 0.25 | 0.33 | 0.40 | 0.55 | 0.63 | 0.12 | 0.38 |
| | SVD–LLM* | 9.27 | 15.63 | 0.29 | 0.59 | 0.52 | 0.68 | 0.69 | 0.20 | 0.49 |
| | **DF-SVD** | **12.32** | **41.91** | **0.26** | **0.48** | **0.43** | **0.61** | **0.63** | **0.22** | **0.44** |
| | **DF-SVD*** | **8.96** | **14.23** | 0.27 | **0.62** | 0.47 | 0.64 | **0.70** | **0.24** | **0.49** |
| 50% | SVD | 131715 | 79815 | 0.16 | 0.26 | 0.26 | 0.50 | 0.52 | 0.19 | 0.31 |
| | FWSVD | 24391 | 23104 | 0.12 | 0.26 | 0.26 | 0.50 | 0.53 | 0.20 | 0.31 |
| | ASVD | 15358 | 27929 | 0.12 | 0.26 | 0.26 | 0.51 | 0.52 | 0.19 | 0.31 |
| | SVD-LLM | 23.97 | 118.57 | 0.16 | 0.33 | 0.29 | 0.54 | 0.56 | 0.21 | 0.34 |
| | SVD–LLM* | 15.30 | 19.26 | 0.22 | 0.54 | 0.40 | 0.59 | 0.67 | 0.23 | 0.44 |
| | **DF-SVD** | **21.85** | **107.51** | **0.17** | **0.38** | **0.30** | **0.56** | **0.57** | **0.22** | **0.37** |
| | **DF-SVD*** | **10.79** | **17.41** | **0.22** | **0.55** | **0.41** | **0.59** | **0.67** | **0.23** | **0.45** |
| 60% | SVD | 105474 | 106976 | 0.16 | 0.26 | 0.26 | 0.50 | 0.52 | 0.21 | 0.31 |
| | FWSVD | 32194 | 29292 | 0.15 | 0.26 | 0.26 | 0.50 | 0.53 | 0.18 | 0.31 |
| | ASVD | 57057 | 43036 | 0.12 | 0.26 | 0.26 | 0.49 | 0.51 | 0.18 | 0.30 |
| | SVD-LLM | 66.62 | 471.83 | 0.10 | 0.05 | 0.10 | 0.17 | 0.21 | 0.04 | 0.11 |
| | SVD–LLM* | 15.00 | 26.26 | 0.18 | 0.42 | 0.31 | 0.44 | 0.35 | 0.12 | 0.30 |
| | **DF-SVD** | **55.27** | **289.87** | **0.13** | **0.30** | **0.27** | **0.51** | **0.54** | **0.21** | **0.32** |
| | **DF-SVD*** | **13.33** | **21.02** | **0.21** | **0.53** | **0.38** | **0.56** | **0.66** | **0.23** | **0.44** |

**Comparison of Efficiency.** We compare the compression time of DF-SVD, SVD-LLM, and ASVD on LLaMA-7B at a 40% compression ratio. As shown in Table 2, DF-SVD completes the end-to-end compression pipeline significantly faster than both SVD-LLM and ASVD, highlighting its practical efficiency.

**Comparison with Dobi-SVD and AdaSVD.** We further compare our decay-aware rank allocation with Dobi-SVD Qinsi et al. (2025) and AdaSVD Li et al. (2025), which also dynamically allocate ranks. For fairness, we use the basic Dobi-SVD variant with rank search and weight updating only. As shown in Table 3, DF-SVD achieves higher accuracy while being more efficient.

**Performance across Different LLMs.** To examine the generability of DF-SVD across different LLMs, we compare it with SVD-based baselines on LLaMA and OPT models at a 30% compression ratio on WikiText-2. As shown in Table 4, DF-SVD consistently outperforms vanilla SVD, FWSVD, ASVD, and SVD-LLM, delivering more stable performance across models with much lower compression cost.

**Comparison with Pruning Methods.** We compare the performance of DF-SVD with three state-of-the-art structured pruning-based Large Language Model compression methods: LLM-Pruner Ma et al. (2024), SliceGPT Ashkboos et al. (2024), and BlockPrune Lagunas et al. (2021b), under LLAMA-7B for 20% and 40% model compression rate. Results in Table 5 shows DF-SVD achieves lower perplexity and is substantially faster (approximately 6-24× speedup), demonstrating clear advantages over pruning-based approaches.

**Combination with Quantization Methods.** We combine DF-SVD with quantization methods. As demonstrated in Table 6, DF-SVD with U8V8Σ16 quantization (8-bit for singular vectors, 16-bit for singular values) achieves a lower perplexity with an approximate compression rate, indicating that the integration of DF-SVD and quantization achieves superior performance than current leading quantization methods RTN Nagel et al. (2020) and GPTQ Frantar et al. (2022).

Table 2: Time of DF-SVD, SVD-LLM and ASVD on LLaMA-7B under 40% compression ratio.

| DF-SVD | | | SVD-LLM | | | ASVD | | |
|---|---|---|---|---|---|---|---|---|
| Truncation | Parameter Update | **Total** | Truncation | Parameter Update | **Total** | Normalize | Search | **Total** |
| 10min | **20min** | **30min** | 10min | 3.5h | 3.5h | 5min | 5.5h | 5.5h |

Table 3: Compared to Dobi-SVD and AdaSVD on Llama-7B at 40% Compression Ratio.

| Method | Perplexity↓ | Accuracy↑ |
|---|---|---|
| Original | 5.68 | 0.53 |
| Ada-SVD (2.5 hours) | 14.76 | 0.42 |
| Dobi-SVD (8 hours) | 13.54 | 0.44 |
| **DF-SVD (10 min)** | **12.32** | **0.44** |
| **DF-SVD* (30 min)** | **8.96** | **0.49** |

Table 4: Perplexity of LLAMA2-7B, OPT 6.7B and LLAMA3-8B under 20% compression ratio.

| Method | Llama-2 7B | OPT 6.7B | Llama-3 8B |
|---|---|---|---|
| FWSVD | 2360 | 14559 | 4782 |
| ASVD (5.5 hours) | 10.10 | 82.00 | 17.55 |
| SVD-LLM (10 min) | 8.50 | 16.04 | 14.41 |
| SVD-LLM* (3.5 hours) | 7.73 | 14.47 | 11.41 |
| **DF-SVD (10 min)** | **7.85** | **15.67** | **14.19** |
| **DF-SVD* (30 min)** | **7.12** | **13.48** | **10.30** |

## 4.2 ABLATION STUDY

We conduct an ablation study to assess the contributions of Decay-Aware Rank Allocation and Feature-Preserved Weight Update. DF-SVD-1 uses only the former, DF-SVD-2 only the latter, DF-

SVD-3 combines both without dynamic rank updates, and DF-SVD-4 includes all components. As shown in Table 7, all DF-SVD variants on LLaMA-7B at 30% and 50% compression consistently outperform existing methods within 30 minutes.

Table 5: Perplexity of LLAMA-7B compressed by structured pruning methods and DF-SVD.

| Ratio | Model | WikiText2 |
|---|---|---|
| 20% Compression | LLM-Pruner (3 hours) | 9.88 |
| | SliceGPT (3.5 hours) | 8.78 |
| | BlockPrune (12 hours) | 9.4 |
| | **DF-SVD (30 min)** | **7.24** |
| 40% Compression | LLM-Pruner (3 hours) | 18.94 |
| | SliceGPT (3.5 hours) | 16.39 |
| | BlockPrune (12 hours) | 19.78 |
| | **DF-SVD (30 min)** | **8.96** |

Table 6: Perplexity and Memory of LLAMA-7B compressed by Mixed-Precision Quantization.

| Method | Memory | WikiText2 |
|---|---|---|
| RTN (3bit) | 2.80GB | 25.54 |
| GPTQ (3bit) | 2.86GB | 16.65 |
| **DF-SVD (U8V8$\Sigma$16)** | **2.88GB** | **13.33** |

Table 7: Ablation study on Llama-7B at 30% and 50% Compression Ratio.

| Method | 30% Compression Ratio | | 50% Compression Ratio | |
|---|---|---|---|---|
| | Perplexity↓ | Accuracy↑ | Perplexity↓ | Accuracy↑ |
| ASVD (5.5 hours) | 51 | 0.39 | 15358 | 0.31 |
| SVD–LLM (10 min) | 9.56 | 0.42 | 23.97 | 0.34 |
| SVD–LLM* (3.5 hours) | 8.13 | 0.50 | 15.30 | 0.44 |
| **DF-SVD-1 (10 min)** | **9.05** | **0.46** | **21.85** | **0.37** |
| **DF-SVD-2 (30 min)** | **8.01** | **0.49** | **12.39** | **0.42** |
| **DF-SVD-3 (30 min)** | **7.92** | **0.50** | **11.46** | **0.43** |
| **DF-SVD-4 (30 min)** | **7.87** | **0.51** | **10.79** | **0.45** |

## 5 CONCLUSION

DF-SVD presents a novel framework for accurate and efficient LLM compression via singular value decomposition, addressing two critical limitations of existing SVD-based methods: efficiently determining the optimal truncation and update rank and ensuring high accuracy restoration. By introducing Decay-Aware Rank Allocation, which dynamically assigns truncation and update ranks based on singular value decay characteristics, DF-SVD optimizes layer-specific and weight matrices-specific truncation while minimizing computational overhead. The Feature-Preserved Weight Update method further guarantees high accuracy restoration and fast convergence by fixing critical components and selectively updating minor components, preserving pre-trained features and ensuring isotropic Hessian properties. Experimental results demonstrate significant improvements: (1) Superior performance across LLAMA, LLAMA2, LLAMA3 and OPT models at 30–60% compression ratios; (2) 7–16× faster compression compared to SVD-LLM, ASVD, and Dobi-SVD; (3) Consistent outperformance against pruning and quantization baselines while maintaining efficiency and accuracy. Through these innovations, DF-SVD establishes a direct link between singular spectrum characteristics and training-free rank selection while enhancing Hessian isotropy, ultimately advancing a novel paradigm for accurate and efficient SVD-based LLM compression.

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

# A APPENDIX

## A.1 EMPIRICAL VALIDATION OF THE EXPONENTIAL SPECTRAL DECAY ASSUMPTION

Our analysis in this section is based on the assumption that the singular values of transformer weight matrices follow an approximately exponential decay,

$$\sigma_i \approx \sigma_0 \exp(-\lambda i).$$

To ensure that this assumption reflects actual model behavior, rather than a modeling convenience, we conduct a comprehensive empirical study on all projection matrices subject to low-rank approximation in our method. Specifically, we examine the four attention projections (Q, K, V, O) and the three MLP projections (gate, up, down) across all 32 layers of LLaMA-7b, yielding 224 matrices in total. For each matrix $W$, we extract singular values via SVD and normalize them by the top singular value using $\tilde{\sigma}_i = \sigma_i/\sigma_0$.

To assess which parametric family best captures the observed decay behavior, we fit six commonly used monotonic decay models:

Exponential model is defined by the equation:

$$\tilde{\sigma}_i = \exp(-\lambda i),$$

Logarithmic model takes the form:

$$\tilde{\sigma}_i = \max(0,\ a - b\log(i+1)),$$

Power-law model is given by:

$$\tilde{\sigma}_i = c\,i^{-\alpha},$$

Linear decay model is expressed as:

$$\tilde{\sigma}_i = \max(0,\ a - bi),$$

Quadratic polynomial model follows:

$$\tilde{\sigma}_i = a_2 i^2 + a_1 i + a_0,$$

and Hyperbolic model is described by:

$$\tilde{\sigma}_i = \frac{a}{b+i}.$$

All parameters are estimated in log-space using Huber regression to mitigate the effect of large residuals. For models that can produce non-positive predictions, we impose non-negativity before evaluating errors. We also validate the correctness of each fitting routine using synthetic sequences generated exactly from the corresponding analytic forms, which consistently produce near-perfect fits (with $R^2 > 0.999$).

To compare the expressive accuracy of these decay families, we report four metrics: $R^2$ computed in log-space, and MAE, RMSE, and MAPE computed on the original scale. The average results across all 224 matrices are summarized in Table 8 and visualized in Figure 5. Although logarithmic and power-law models can achieve relatively high $R^2$ on some matrices, they consistently exhibit larger MAE, RMSE, and especially MAPE, indicating noticeably worse accuracy on the original singular-value scale.Similar trends are observed when we restrict the analysis to attention and MLP projection matrices, as shown in Figures 6–7, where the exponential family remains the most accurate choice across modules and layers. Overall, the exponential model achieves the best absolute accuracy, making it the most suitable approximation for energy-based rank allocation.

In contrast, the exponential model achieves the best absolute accuracy overall, making it the most suitable approximation for energy-based rank allocation. Together, these results demonstrate that among six decay families, exponential decay provides the most accurate and stable approximation of real singular-value spectra, with consistent behavior across modules and layers. This empirical evidence supports adopting the exponential spectral prior in the theoretical analysis presented in Appendix A.2.

Figure 5: Overall comparison of singular value decay models.

|  | **Exp** | **Poly** | **Log** | **Hyp** | **Lin** | **Pow** |
|---|---|---|---|---|---|---|
| **Overall (224 matrices)** | | | | | | |
| $R^2$ | **0.9435** | 0.9278 | 0.9292 | 0.9261 | 0.8751 | 0.7377 |
| MAE | **0.012585** | 0.011794 | 0.021157 | 0.021289 | 0.018761 | 0.046543 |
| RMSE | **0.021829** | 0.022558 | 0.027485 | 0.028007 | 0.031224 | 0.055947 |
| MAPE | **136.22** | 349.22 | 358.99 | 387.18 | 483.38 | 698.50 |
| **Attention Modules (128 matrices)** | | | | | | |
| $R^2$ | **0.9471** | 0.9230 | 0.9350 | 0.9174 | 0.8517 | 0.7055 |
| MAE | **0.013580** | 0.011964 | 0.020309 | 0.025378 | 0.021731 | 0.051957 |
| RMSE | **0.022171** | 0.022843 | 0.026885 | 0.031563 | 0.034406 | 0.061898 |
| MAPE | **234.37** | 606.90 | 619.78 | 671.09 | 840.89 | 1206.25 |
| **MLP Modules (96 matrices)** | | | | | | |
| $R^2$ | **0.9388** | 0.9342 | 0.9216 | 0.9375 | 0.9063 | 0.7807 |
| MAE | **0.011260** | 0.011567 | 0.022289 | 0.015837 | 0.014801 | 0.039325 |
| RMSE | **0.021373** | 0.022177 | 0.028285 | 0.023265 | 0.026980 | 0.048013 |
| MAPE | **5.35** | 5.64 | 11.28 | 8.63 | 6.71 | 21.51 |

Table 8: Unified comparison of six decay models over overall, attention, MLP projection matrices.

## A.2 MATHEMATICAL DERIVATION: EXPONENTIAL DECAY OF SINGULAR VALUE SPECTRUM IN LLMS

Let $W \in \mathbb{R}^{m \times n}$ be a linear weight matrix in a large language model (LLM), with singular value decomposition (SVD)

$$W = U\Sigma V^\top,$$

where the singular values satisfy

$$\sigma_1 \geq \sigma_2 \geq \cdots \geq \sigma_r \geq 0, \quad r = \min(m, n).$$

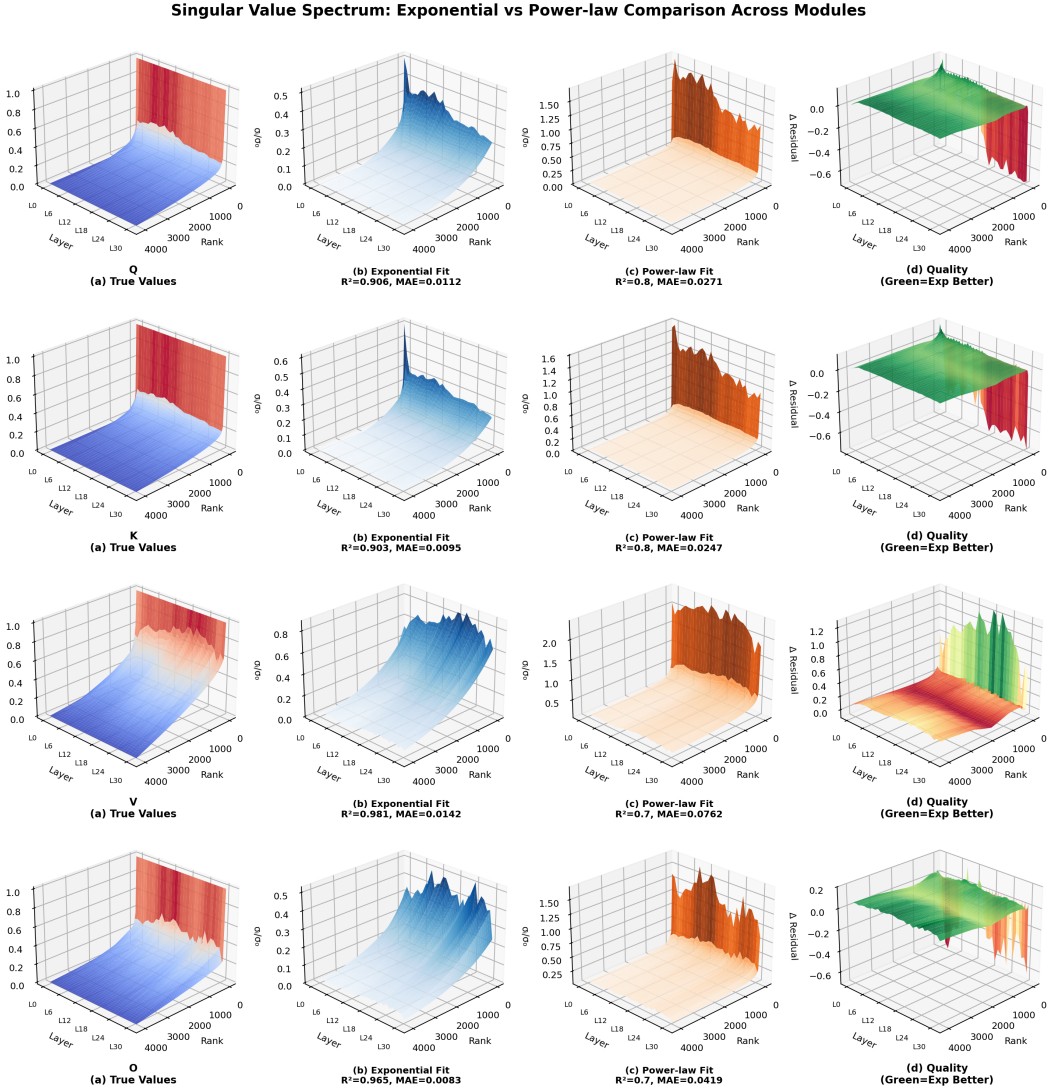

Figure 6: Overall comparison of decay models on Attention projections.

We aim to establish that, under appropriate structural and distributional assumptions, there exist constants $C > 0$ and $\lambda > 0$ such that

$$\sigma_k(W) \le C\, e^{-\lambda k}, \quad \forall k \in \{1, \ldots, r\},$$

meaning that the singular value spectrum follows an exponential decay model. Consider modeling certain LLM mappings, including attention weight matrices, as discretizations of compact integral operators generated by analytic kernels. Let

$$(Tf)(x) = \int_{\Omega_y} K(x, y)\, f(y)\, dy$$

be a compact operator $T : L^2(\Omega_y) \to L^2(\Omega_x)$, where $\Omega_x, \Omega_y \subset \mathbb{R}$ are bounded domains, and the kernel $K(x, y)$ is analytic and bounded in a complex neighborhood $\mathcal{D}_x \supset \Omega_x$, $\mathcal{D}_y \supset \Omega_y$. Analyticity guarantees an orthogonal basis expansion

$$K(x, y) = \sum_{i=0}^{\infty} \sum_{j=0}^{\infty} c_{ij}\, p_i(x)\, q_j(y),$$

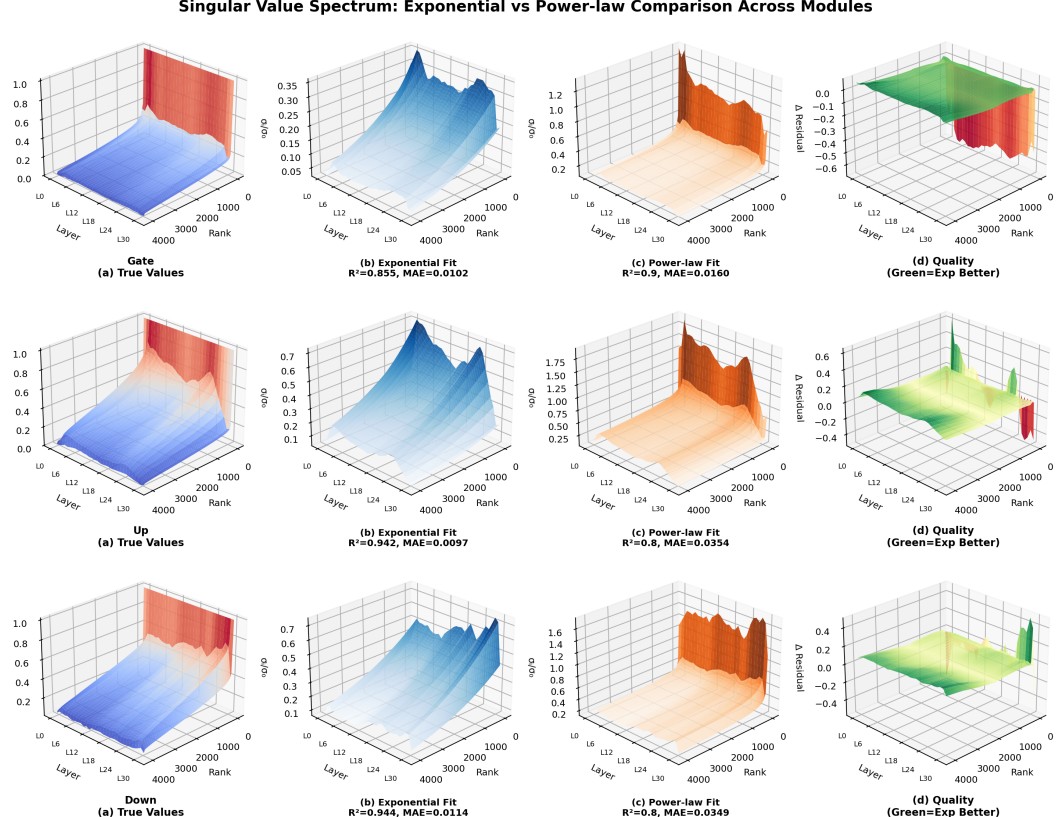

Figure 7: Overall comparison of decay models on MLP projections.

where $\{p_i\}$ and $\{q_j\}$ are orthogonal basis functions (e.g., Chebyshev or Legendre polynomials), and there exist constants $M > 0$ and $\rho \in (0, 1)$ such that

$$|c_{ij}| \leq M\rho^{i+j}.$$

This geometric decay in coefficients follows from standard approximation theory for analytic functions, via the maximum modulus principle and Jackson-type inequalities. Now truncate the expansion to total degree $k$:

$$K_k(x, y) = \sum_{i+j<k} c_{ij}\, p_i(x)\, q_j(y),$$

which induces a rank-$k$ operator $T_k$. The truncation error satisfies

$$\|T - T_k\|_{L^2 \to L^2} \leq C_1\, \rho^{\,k},$$

since the discarded coefficients form the tail of a convergent double geometric series. By the definition of approximation numbers $a_k(T)$,

$$a_k(T) := \inf_{\mathrm{rank}(A)<k} \|T - A\|_{L^2 \to L^2},$$

and by the equality $a_k(T) = s_k(T)$ for compact operators (where $s_k(T)$ denotes the $k$-th singular value), we have

$$s_k(T) \leq C_1\, \rho^{\,k} = C\, e^{-\lambda k}, \qquad \lambda := -\ln \rho > 0.$$

Thus, analytic kernels yield integral operators whose singular values decay exponentially. When discretizing $T$ into $W_N \in \mathbb{R}^{N_x \times N_y}$ via stable Nyström or Galerkin schemes with quadrature weights $\omega_j$,

$$(W_N)_{ij} \approx K(x_i, y_j)\, \omega_j,$$

spectral approximation theory ensures that for sufficiently large $N$, the singular values $\sigma_k(W_N)$ approximate $s_k(T)$ for $k \leq \kappa N$ with small uniform error. Consequently,

$$\sigma_k(W_N) \leq C'e^{-\lambda' k}, \quad 1 \leq k \leq \kappa N$$

for constants $C', \lambda' > 0$. This framework applies naturally to LLM architectures. The self-attention mechanism produces a weight matrix

$$A(X) = \text{Softmax}\left(\frac{Q(X)K(X)^\top}{\sqrt{d}}\right),$$

where $Q(X) = XW_Q$ and $K(X) = XW_K$. Given that $Q$, $K$ and the Softmax function are analytic mappings on the relevant data manifold, the attention kernel $K_{\text{attn}}(i, j)$ varies smoothly with token positions and satisfies the analyticity assumption above, leading to exponential decay in its singular value spectrum. Similarly, feed-forward layers can be modeled as compositions $W_2 \, \phi \, W_1$, with $\phi$ an analytic activation. If the input covariance spectrum itself decays exponentially,

$$\lambda_k(\Sigma_x) \leq C_x e^{-\alpha k},$$

and $W$ is bounded in the corresponding eigenbasis, then

$$\sigma_k(W) \leq \|W\|\sqrt{\lambda_k(\Sigma_x)} \leq \|W\|\sqrt{C_x}\, e^{-\frac{\alpha}{2}k},$$

implying again exponential singular value decay. Under the combined conditions of analytic kernel structure, stable discretization, and compressible (exponentially decaying) data covariance spectrum, it follows rigorously from approximation theory that there exist constants $C, \lambda > 0$ such that

$$\sigma_k(W) \leq Ce^{-\lambda k}, \quad k = 1, \ldots, r.$$

Moreover, in a semilog plot of $\log \sigma_k$ versus $k$, the points lie approximately on a straight line with slope $-\lambda$, which matches empirical observations for trained LLM weight matrices.

## A.3 PERFORMANCE ON LLMS WITH LARGER SCALES.

To evaluate DF-SVD's generalizability across LLMs of varying scales, we compared its performance with baseline methods on LLaMA 13B and 30B at a 20% compression ratio, and additionally evaluate LLaMA2-70B under a 30% compression ratio. As shown in Table 9, DF-SVD maintains strong performance and consistently outperforms SVD-based baselines, indicating that its decomposition strategy scales favorably with model size and aligns well with the scaling laws observed in large language models.

Table 9: Performance of LLaMA-13B, LLaMA-30B and under 20% compression ratio

| Method | LLaMA-13B | | LLaMA-30B | |
|---|---|---|---|---|
| | Perplexity ↓ | Accuracy ↑ | Perplexity ↓ | Accuracy ↑ |
| Original | 5.09 | 0.59 | 4.10 | 0.61 |
| SVD-LLM | 6.61 | 0.54 | 5.63 | 0.57 |
| SVD-LLM* | 6.43 | 0.55 | 5.14 | 0.59 |
| **DF-SVD** | **6.32** | **0.55** | **5.25** | **0.58** |
| **DF-SVD*** | **6.08** | **0.57** | **5.02** | **0.60** |

## A.4 THE USE OF LARGE LANGUAGE MODELS

We have utilized a Large Language Model (LLM) to assist in refining the language and improving the clarity of the manuscript. The LLM was primarily used for enhancing the overall phrasing and readability of the text, ensuring a more polished and professional presentation.

Table 10: Performance of LLaMA2-70B under 30% compression ratio

| Method | LLaMA2-70B | |
|---|---|---|
| | Perplexity ↓ | Accuracy ↑ |
| Original | 3.32 | 0.74 |
| SVD-LLM | 6.74 | 0.58 |
| SVD-LLM* | 5.44 | 0.65 |
| **DF-SVD** | **5.69** | **0.63** |
| **DF-SVD*** | **4.97** | **0.68** |

### A.5 SPEEDUP ANALYSIS ACROSS COMPRESSION RATIOS

To verify that DF-SVD can enhance inference efficiency on real hardware, we measure the numbers of tokens that the original LLaMA-7B and its compressed version by DF-SVD generate per second with different batch sizes and sequence lengths on both GPU and CPU. The results in Figure A.5 demonstrate that DF-SVD consistently boosts generation speed across all compression ratios. Notably, the improvement becomes more pronounced as the batch size grows and the sequence length shortens.

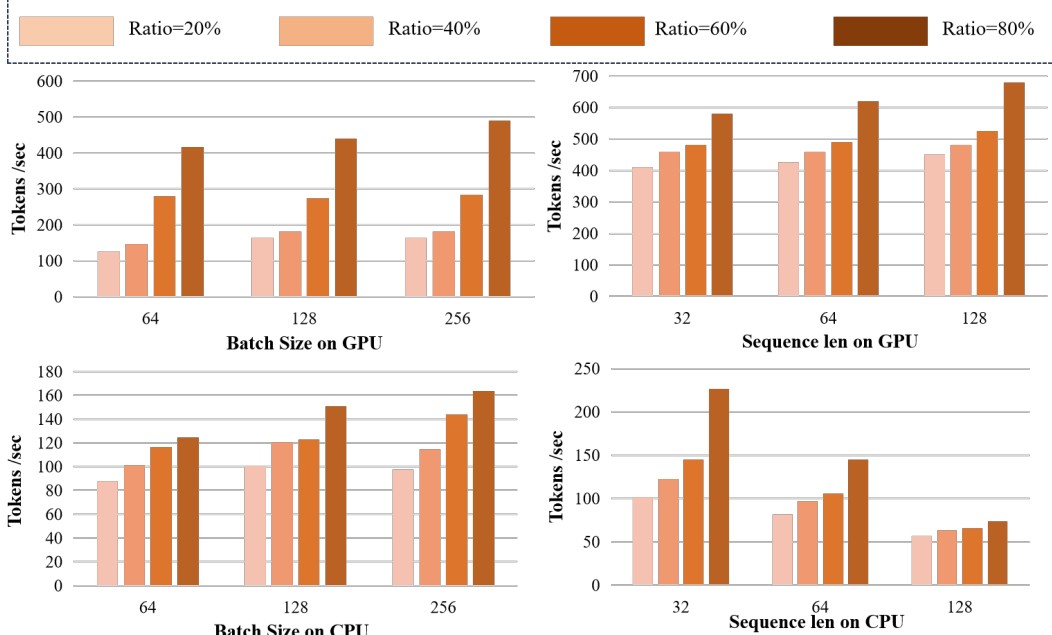

Figure 8: Throughout (Tokens/sec) of original LLAMA-7B and its computed version by DF-SVD under 20%, 40%, 60% and 80% compression ration on single A800 GPU (Figure (a), (b)) and single Intel(R) Xeon(R) Platinum 8378A CPU (Figure (c), (d)).

### A.6 ALGORITHMS

Algorithm 1 shows the pseudocode of DF-SVD. Before compression, DF-SVD randomly collects a small amount of sentences as the calibration data $C$, it then fetches whitening matrix and runs the Decay-Aware Rank Allocation process as shown in Algorithm 2 to obtain the set of Feature-Preserved Weight Update. Instead of directly finishing the whole compression, it stores the decomposed matrices and further utilizes these matrices to run the Feature-Preserved Weight Update as shown in Algorithm 3.

---

**Algorithm 1** Pseudocode of DF-SVD

---

1: **Input:** $M$: Original LLM
2: **Output:** $M'$: Compressed LLM by DF-SVD
3: **procedure** DF-SVD($M$)
4:     Randomly collect several sentences as calibration data $C$
5:     $\text{Set}_S \leftarrow \emptyset$ {Initialize the set of whitening matrices}
6:     $\text{Set}_{\text{SVD}} \leftarrow \emptyset$ {Store decomposed matrices}
7:     $\text{Set}_W \leftarrow$ weights in $M$ {All compressible weights}
8:     **for** $W$ in $\text{Set}_W$ **do**
9:         $X \leftarrow M(W, C)$ {Obtain the input activation of the weight matrix $M$}
10:        $S \leftarrow$ Cholesky Decomposition($XX^\top$) {Apply cholesky decomposition on $XX^\top$}
11:        $\text{Set}_S \leftarrow S \cup \text{Set}_S$ {Store the whitening weight matrix in the set}
12:        $S \leftarrow \text{Set}_S(W)$ {Fetch whitening matrix}
13:        $U\Sigma, V^\top \leftarrow$ **Decay-Aware Rank Allocation**($U\Sigma, V$)
14:        $\text{Set}_{\text{SVD}} \leftarrow (U\Sigma, V^\top) \cup \text{Set}_{\text{SVD}}$ {Store matrices}
15:        $W_u \leftarrow U\Sigma$
16:        $W_v \leftarrow V^\top S^{-1}$
17:     **end for**
18:     $M' \leftarrow$ **Feature-Preserved Weight Update**($M, C, \text{Set}_S, \text{Set}_{\text{SVD}}$)
19:     **return** $M'$
20: **end procedure**

---

**Algorithm 2** Pseudocode of Decay-Aware Rank Allocation

---

1: **Input:** Original singular value matrix
2: **Output:** $U\Sigma, V^\top$: singular value matrix after truncation
3: **procedure** Decay-Aware Rank Allocation($U\Sigma, V^\top$)
4:     $\text{Set}_W \leftarrow$ weights in $M$ {All compressible weights}
5:     **for** $W$ in $\text{Set}_W$ **do**
6:         $\sigma_i = \sigma_0 e^{-\lambda i}$ {Estimate decay parameters $(\sigma_0, \lambda)$ using nonlinear regression}
7:         $\lambda_{\text{norm}} \leftarrow \frac{\lambda - \min(\lambda)}{\max(\lambda) - \min(\lambda)}$ {Get the normalized decay coefficient}
8:         $ra_{\text{trunc}} = ra_{\text{old}} * (1 - \lambda_{\text{norm}})$ {Get the adjusted reserve ratio}
9:         $r_{\text{update}} = \text{rank}_{\text{old}} * (1 - \lambda_{\text{norm}})$ {Get the new updating rank}
10:        Dynamically adjust the compression ratio in the truncation process based on $ra_{\text{trunc}}$, as well as dynamically adjust the rank in the weight updating process based on $r_{update}$
11:     **end for**
12:     **return** new truncated matrix $U\Sigma, V^\top$
13: **end procedure**

---

---

**Algorithm 3** Pseudocode of Feature-Preserved Parameter Update

---

1: **Input:** $M$: Original LLM
2: **Input:** $C$: Calibration Data
3: **Input:** $\text{Set}_S$: Set of whitening matrices for the weight to compress in $M$
4: **Input:** $\text{Set}_{SVD}$: Set of decomposed matrices for the weight to compress in $M$
5: **Output:** $M'$: Compressed LLM by DF-SVD
6: **procedure** Feature-Preserved Parameter Update $(M, C, \text{Set}_S, \text{Set}_{SVD})$
7:      $M' \leftarrow M$ {Initialize $M'$ with $M$}
8:      $\text{Set}_L \leftarrow M'$ {Obtain the set of encoder and decoder layers in $M'$}
9:      $X' \leftarrow M'(C)$ {Obtain the input activation of the first layer in $M'$}
10:      **for** $L$ in $\text{Set}_L$ **do**
11:          $\text{Set}_W \leftarrow L$ {Obtain the set of weights in $L$ to compress}
12:          **for** $W$ in $\text{Set}_W$ **do**
13:              $S \leftarrow \text{Set}_S(W)$ {Extract the whitening matrix of current weight $W$}
14:              $U\Sigma, V^\top S^{-1} \leftarrow$ Decay-Aware Rank Allocation$(W)$ {Obtain the decomposed matrices of $W$, fix $V^\top S^{-1}$, jointly optimize matrix $U\Sigma$}
15:              $U', \Sigma', V' \leftarrow \text{SVD}(U\Sigma)$ {Decompose $U\Sigma$}
16:              $U'\Sigma'V'^\top \leftarrow U'_p\Sigma'_pV'^\top_p + U'_m\Sigma'_mV'^\top_m$ {Extract principal components(fixed) and minor components (trainable)}
17:              $B_m, A_m \leftarrow U'_m\sqrt{\Sigma'_m}, \sqrt{\Sigma'_m}V'^\top_m$ {Initialize minor components as low-rank matrices}
18:              Dynamically adjust the training rank of each module determined by $r_{update}$
19:              $L(W) \leftarrow L(W_u, W_v)$ {Replace $W$ with $W_u$ and $W_v$ in $L$}
20:          **end for**
21:          Update layer by layer
22:      **end for**
23:      **return** $M'$
24: **end procedure**

---

