# OpenReview forum: "Accurate and Efficient Singular Value Decomposition For LLMs via Decay-aware Rank Allocation and Feature-Preserved Weight Update"
_ICLR.cc/2026/Conference — Submitted to ICLR 2026_

### Official Review · Reviewer_LAV4 · 2025-10-30

**Soundness:** 3
**Presentation:** 3
**Contribution:** 3
**Rating:** 6
**Confidence:** 4

**Summary:**

This paper proposes DF-SVD, a method for compressing LLM using SVD with two main contributions: (1) decay-aware rank allocation that dynamically determines truncation and update ranks based on singular value decay, and (2) feature-preserved weight updates that achieve isotropic Hessian by fixing V^TS^{-1} and selectively updating only minor components of UΣ. The method achieves much speedup over existing SVD-based methods while maintaining or improving accuracy.

**Strengths:**

1. Strong exp results and practical speedup.
2. Sound theoretical analysis: The Hessian conditioning analysis (section 3.2) is mathematically sound and provides clear intuition for why the proposed reformulation can achieve better convergence properties

**Weaknesses:**

1. Limited novelty relative to SVD-LLM: The paper heavily builds on SVD-LLM's foundation (Cholesky whitening, sequential optimization framework, experimental setup). Much of the methodology is inherited, making this more of an incremental improvement.
2. Missing critical comparisons:
A. No comparison with AdaLoRA: The paper cites AdaLoRA for importance-based rank allocation but never compares against it
B. No comparison with other methods: Methods like "Dynamic Low-rank Estimation for Transformer-based Language Models" (Hua et al., EMNLP 2023 findings) are highly relevant but not discussed or compared
3. No empirical validation that decay coefficient actually correlates with ground-truth importance (e.g., gradient magnitudes, ablation impact)

**Questions:**

1. Can you show via experiments that Hessian isotropy causes the speedup (e.g., via iteration counts, convergence curves)?
2. Why not compare with AdaLoRA, which you cite as inspiration?
3. What is the correlation between λ_norm and ground-truth importance metrics (gradients, sensitivity)?

---

> ### Author Response · Authors · 2025-11-19
> **Response to Reviewer LAV4**
>
> We sincerely thank the reviewer for the insightful and technically deep comments. In the revised manuscript, we have added convergence-curve experiments to empirically verify the link between Hessian isotropy and optimization speed, clarified our conceptual relationship to AdaLoRA and why we focus comparisons on post-training SVD-based compression methods, and elaborated on how the normalized decay coefficient λ_norm acts as a training-free proxy for weight importance. All corresponding additions and clarifications have been incorporated into the main text and appendix and are highlighted in blue for your convenience.
>
> **Response to weakness 1:**
>
> We thank the reviewer for this insightful question. According to your suggestion, we have added an experiment to directly examine the effect of Hessian isotropy on optimization speed in the revised version of the paper. Specifically, we compare DF-SVD with SVD-LLM on the reconstruction fine-tuning of LLaMA-7B. As shown in the newly added convergence curves in Figure 4, DF-SVD converges in fewer iterations, maintains consistently lower training loss, and exhibits a clear performance gap over SVD-LLM throughout the optimization process.
>
> These empirical results are fully consistent with our Hessian analysis: enforcing Hessian isotropy leads to more stable and effective weight updates, which in turn accelerates convergence and improves final performance. We have updated the manuscript accordingly to include these curves and the accompanying discussion.
>
> **Response to weakness 2:**
>
> Thank you very much for this insightful suggestion. We indeed draw conceptual inspiration from AdaLoRA, particularly its use of spectrum-aware adaptive rank allocation.
>
> However, our work is positioned in a somewhat different setting: we focus on post-training SVD-based compression of the base model, whereas AdaLoRA is a parameter-efficient fine-tuning (PEFT) method that introduces additional low-rank adaptation modules and relies on task-specific training data and fine-tuning. Because of this difference in problem setting and objectives, our experimental comparison is mainly aligned with post-training compression baselines under the same assumptions and evaluation protocol (e.g., SVD-LLM, AdaSVD, Dobi-SVD, FWSVD, ASVD, as well as pruning and quantization methods).
>
> We appreciate the reviewer’s suggestion and will revise the related work section to more clearly explain our relationship to AdaLoRA: we build on a similar idea of adaptive rank allocation at the conceptual level, but our method is designed specifically for SVD-based post-training compression, and thus we focus our empirical comparisons on methods in this particular setting.
>
> **Response to weakness 3:**
>
> The correlation between λ_norm and ground-truth importance metrics (e.g., gradients and sensitivity) naturally follows from our decay-aware rank allocation framework. A lower λ_norm (slow spectral decay) corresponds to matrices that encode more critical features, exhibit larger gradients, and are more sensitive to perturbations, whereas a higher λ_norm (rapid decay) is associated with smaller gradients and lower sensitivity, indicating lower importance.
>
> In our decay-aware rank allocation, matrices with low λ_norm are assigned higher effective ranks and undergo less aggressive compression, while those with high λ_norm are compressed more aggressively. Consequently, λ_norm serves as a training-free proxy for weight importance, guiding the preservation of weights associated with slow-decaying singular values, which are most crucial for maintaining accuracy.

---

> ### Author Response · Authors · 2025-11-26
>
> Dear Reviewer,
>
> I hope this message finds you well. As the discussion period is nearing its end, I wanted to ensure we have addressed all your concerns satisfactorily. If there are any additional points or feedback you'd like us to consider, please let us know. Your insights are invaluable to us, and we’re eager to address any remaining issues to improve our work.
>
> Thank you for your time and effort in reviewing our paper!

---

### Official Review · Reviewer_y1EY · 2025-10-31

**Soundness:** 3
**Presentation:** 3
**Contribution:** 3
**Rating:** 4
**Confidence:** 4

**Summary:**

This paper introduces the DF-SVD framework, which aims to simultaneously improve accuracy recovery and compression efficiency in large language models. The authors first identify two key issues with conventional SVD-based compression methods: difficulty in selecting appropriate truncation and update ranks, and limited fine-tuning stability. To address these challenges, the paper proposes two core modules. The first, Decay-Aware Rank Allocation, models the singular value decay rate of each layer’s weight matrix to dynamically determine both truncation and update ranks, achieving adaptive compression across layers and matrices. The second, Feature-Preserved Weight Update, freezes the dominant components and only updates the minor subspace, thereby preserving critical pretrained features while improving the isotropy of the Hessian for faster convergence. Experimental results show that DF-SVD consistently outperforms existing methods such as SVD-LLM, ASVD, and Dobi-SVD on LLaMA, LLaMA2, LLaMA3, and OPT models under 30–60% compression, achieving comparable accuracy with 7–16× faster end-to-end compression.

**Strengths:**

1. Clear and practical implementation design.
The paper follows the SVD-LLM pipeline with whitening (via Cholesky decomposition) and SVD pre-processing, while confining its innovation to rank allocation and the update subspace. This design choice makes the method easy to reproduce, integrate, and deploy in real-world model compression workflows.

2. Comprehensive experimental evaluation.
The experiments cover multiple model families (LLaMA and OPT) and diverse datasets, and report both accuracy and end-to-end compression time. The study also compares DF-SVD against pruning and quantization methods, demonstrating its compatibility and potential for combined use.

**Weaknesses:**

1. Limited novelty.
The paper’s motivation—improving rank selection and reducing update time—targets a well-studied problem. While the proposed approach is practical, it appears relatively straightforward and lacks deeper theoretical innovation. For example, using singular value decay as a heuristic for rank allocation is intuitive but overlooks inter-layer importance differences; in practice, some critical layers may still require higher ranks even with rapid singular value decay.

2. Insufficient validation of the exponential decay assumption.
The core rank allocation mechanism hinges on the assumption that singular values follow an approximately exponential decay pattern and can be modeled by a single parameter λ. Although the paper provides preliminary theoretical reasoning and empirical evidence, it lacks sensitivity analyses showing how deviations from this assumption affect model performance, as well as more rigorous theoretical justification.

3. Under-examined assumptions in the optimization analysis.
The theoretical claim that the Hessian becomes isotropic (𝐻=2𝐼) depends on the assumption of nearly orthogonal, whitened inputs. However, it remains unclear whether this assumption holds under small-sample calibration or distributional shift, and whether it is consistent across different layers or batches. While freezing principal components may preserve pretrained knowledge, it could hinder adaptation in cases of aggressive compression or significant domain shift.

4. Modest empirical gains.
In the reported results, DF-SVD achieves only marginal improvements over SVD-LLM in accuracy, which may not be sufficient to demonstrate a strong advantage given that DF-SVD employs mixed-rank allocation whereas SVD-LLM uses a fixed rank. Although DF-SVD shows faster compression compared with Ada-SVD and Dobi-SVD, the performance comparisons are not exhaustive, leaving some uncertainty about its overall effectiveness.

**Questions:**

Refer to the weakness section

---

> ### Author Response · Authors · 2025-11-19
> **Response to Weakness 1-2**
>
> We sincerely thank the reviewer for the thoughtful and detailed feedback. In the revised manuscript, we have substantially strengthened the theoretical foundations of DF-SVD, provided more rigorous justification and empirical validation for the exponential decay assumption, clarified the scope and robustness of our optimization analysis, and expanded the experimental comparisons to better quantify the empirical gains. All corresponding revisions and newly added discussions have been incorporated into the main text and appendix and are highlighted in blue for your convenience.
>
> **Response to Weakness1:**
>
> Thank you very much for your constructive comments. We have substantially strengthened the theoretical foundations of our method in the revised manuscript. Specifically, we:
>
> Provide experimental and mathematical support for the exponential decay behavior of singular value spectra in LLMs and its implications for rank allocation (Appendix A.1 provides empirical evidence supporting this assumption through comparisons with logarithmic and power-law alternatives, while Appendix A.2 offers theoretical justification grounded in operator theory and approximation theory). This analysis explicitly links the decay rate to the retained energy at a given rank and shows that our decay-aware rank allocation is theoretically superior to fixed-rank strategies under the exponential decay assumption.
>
> Clarify that our decay-aware strategy is inherently designed and applied in a fully module-wise and layer-wise manner. For each weight matrix, we fit an exponential decay model to obtain a matrix-specific decay coefficient λ, normalize it, and use it to jointly adjust truncation and update ranks across all layers and modules. In practice, this decay coefficient naturally encodes feature importance: matrices with slower decay receive higher ranks, while rapidly decaying matrices are more aggressively compressed with minimal loss.
>
> Theoretically justify the proposed Feature-preserved weight update rather than treating it as an ad-hoc engineering heuristic. In Section 3.2, we refine and extend the theoretical derivation to show that, under the original SVD-LLM update scheme, the Hessian eigenvalues scale with the singular values, leading to a potentially large condition number, whereas our reparameterization yields a better-conditioned optimization problem with more stable and faster convergence. Building on this, we further decompose UΣ into principal and minor components, which explains why fixing principal components while updating only minor components both preserves pretrained knowledge and effectively bounds the Hessian condition number during fine-tuning.
>
> We believe that, with the above modifications and clarifications, our method will be more clearly presented as being supported by solid mathematical analysis rather than purely intuitive heuristics.
>
> **Response to Weakness2:**
>
> We thank the reviewer for this helpful comment. Following the reviewer’s suggestion, we now provide both experimental and mathematical support for the exponential decay behavior of the singular value spectra in LLMs and its implications for rank allocation (Appendix A.1 provides empirical evidence supporting this assumption through comparisons with logarithmic and power-law alternatives, while Appendix A.2 offers theoretical justification grounded in operator theory and approximation theory).
>
> Specifically, we model the learned weight matrices and attention kernels in LLMs as discretizations of compact integral operators with analytic kernels. Classical functional analysis then implies that their singular values decay exponentially, and we show that, under stable discretization, the empirical weight matrices inherit this decay rate. The analyticity assumption is supported by the smooth attention mechanism and the bounded, smooth activations in the feed-forward layers, which together provide a principled theoretical explanation for the observed exponential decay rather than treating it as a purely empirical phenomenon.
>
> In addition, several prior studies have reported findings consistent with ours [1][2], further corroborating both the exponential decay assumption and its utility as a principled basis for decay-aware rank allocation in SVD-based LLM compression.
>
> [1] Matthias Thamm, Max Staats, and Bernd Rosenow. Random matrix analysis of deep neural network weight matrices. Physical Review E, 106(5).
>
> [2] Vasiliki Plerou, Parameswaran Gopikrishnan, Bernd Rosenow, Lu´ıs A. Nunes Amaral, Thomas Guhr, and H. Eugene Stanley. Random matrix approach to cross correlations in financial data. Physical Review E, 65(6).

---

> ### Author Response · Authors · 2025-11-19
> **Response to Weakness 3-4**
>
> **Response to Weakness 3:**
>
> Thank you for this insightful comment. We address these concerns both theoretically and empirically as follows:
>
> In DF-SVD, whitening matrices S are obtained via Cholesky decomposition of the empirical covariance XXᵀ computed from calibration data. In practice, this yields S⁻¹X whose covariance is close to identity on the calibration set, but not perfectly orthogonal. Our Hessian derivation therefore applies exactly in the idealized case and approximately in the practical setting. Small calibration sets could yield noisy estimates of XXᵀ and thus imperfect whitening. In our current submission, we already use only several sentences  as calibration data (Algorithm 1, line 4), which is precisely the small-sample regime the reviewer is concerned about.
>
> Our design follows prior findings that large singular directions mainly encode core, task-agnostic semantics, while minor components often capture noisy or long-tail information. DF-SVD therefore fixes the principal components and updates only minor components, which preserves critical pretrained features while still providing a focused adaptation subspace. Moreover, the updating rank r_update is determined by the singular value decay of each weight, so more important weights automatically receive higher updating ranks, enabling effective error correction even at high compression ratios.
>
> We agree that in extreme domain shifts, fully freezing principal components may limit adaptation. Our framework is flexible and can be extended by, for example, lowering the threshold that defines principal singular values so that more components are updated, or assigning a small learning rate to principal components while still concentrating most updates on minor components.
>
> We thank the reviewer again for highlighting these important points. We believe the additional analyses and clarifications will significantly strengthen the rigor and practical relevance of our optimization analysis.
>
> **Response to Weakness 4:**
>
> We thank the reviewer for the constructive comments.
>
> First, regarding the concern about modest empirical gains, our experiments show that DF-SVD almost consistently achieves higher accuracy than existing SVD-based LLM compression methods across a wide range of models, datasets, and compression ratios. This suggests that the proposed mixed-rank strategy provides a robust and practically meaningful improvement rather than a case-specific advantage.
>
> Second, DF-SVD is highly efficient. It is a training-free method that completes the entire compression pipeline within a short wall-clock time, while prior SVD-based approaches often rely on search or heavy optimization and therefore require substantially longer runtimes. This makes DF-SVD particularly suitable for real-world scenarios where both accuracy and turnaround time are critical.
>
> Finally, in our comparisons with Ada-SVD and Dobi-SVD, we evaluate DF-SVD along both accuracy and time dimensions under the same model and compression ratio, and DF-SVD matches or surpasses their accuracy while being significantly faster. We would like to clarify that all reported time costs in these comparisons are end-to-end wall-clock times for the full compression pipeline, rather than partial or per-step timings.
>
> We have made this explicit in experiment section of the revised manuscript to avoid any potential ambiguity.

---

> ### Author Response · Authors · 2025-11-26
>
> Dear Reviewer,
>
> I hope this message finds you well. As the discussion period is nearing its end, I wanted to ensure we have addressed all your concerns satisfactorily. If there are any additional points or feedback you'd like us to consider, please let us know. Your insights are invaluable to us, and we’re eager to address any remaining issues to improve our work.
>
> Thank you for your time and effort in reviewing our paper!

---

### Official Review · Reviewer_CJeg · 2025-11-01

**Soundness:** 2
**Presentation:** 3
**Contribution:** 2
**Rating:** 6
**Confidence:** 2

**Summary:**

This paper proposes DF-SVD, a SVD-based compression framework for large language models (LLMs). It addresses two key challenges in existing SVD compression:
1. Rank Selection Problem – current methods rely on costly search or uniform rank allocation. DF-SVD introduces decay-aware rank allocation, which leverages the singular value spectrum’s decay rate to assign truncation and update ranks per weight matrix dynamically.

2. Limited Accuracy Restoration – sequential weight updates in prior work (e.g., SVD-LLM) lead to Hessian anisotropy and slow convergence. DF-SVD proposes a feature-preserved weight update strategy that freezes principal components and only updates minor components, ensuring Hessian isotropy and preserving pretrained knowledge.

**Strengths:**

1. Clear motivation and problem definition: Identifies two fundamental bottlenecks in SVD compression (rank allocation and update inefficiency).
2. Theoretical contribution: Provides analysis showing Hessian isotropy under the proposed update scheme, linking spectral properties to convergence guarantees.

**Weaknesses:**

1. Generalization to larger models: Experiments are on 7B–8B scale models; it remains uncertain how well DF-SVD scales to 30B+ models.

**Questions:**

Have you tested DF-SVD on huge models (e.g., Qwen3-30B-A3B-Instruct-2507)? Does the efficiency advantage hold at that scale?

---

> ### Author Response · Authors · 2025-11-19
> **Response to Reviewer CJeg**
>
> We sincerely thank the reviewer for raising this important question about scalability to larger models. According to reviewer's suggestion, we have added experiments on larger-scale LLMs. All corresponding additions and clarifications have been incorporated into the revised version and are highlighted in blue for your convenience.
>
> Following the reviewer’s advice, we have added experiments on larger models (LLaMA-13B, LLaMA-30B, and LLaMA2-70B) under the same compression settings, and the corresponding results are summarized in table 1 and table 2 below. These new results show that DF-SVD maintains strong performance and consistently outperforms SVD-based baselines on larger models, indicating that DF-SVD generalizes well to larger models. We have also incorporated the results and the accompanying discussion into section Appendix A.3 of the revised paper.
>
> Table 1 Performance of LLaMA-13B, LLaMA-30B under 20 % compression ratio
> | Method| LLaMA-13B |     | LLaMA-30B |        |
> | :--- | :----| :---| :----| :----|
> |             |Perplexity↓| Accuracy↑| Perplexity↓ | Accuracy↑|
> | Original| 5.09 | 0.59 | 4.10| 0.61|
> |SVD-LLM|6.61|0.54|5.63|0.57|
> |SVD-LLM*|6.43|0.55|5.14|0.59|
> |**DF-SVD**|**6.32**|**0.55**|**5.25**|**0.58**|
> |**DF-SVD***|**6.08**|**0.57**|**5.02**|**0.60**|
>
>
> Table 2 Performance of LLaMA2-70B under 30 % compression ratio
> |  Method| LLaMA2-70B |     |
> | :--- | :----| :---|
> |             |Perplexity↓| Accuracy↑|
> | Original| 3.32 | 0.74 |
> |SVD-LLM|6.74|0.58|
> |SVD-LLM*|5.44|0.65|
> |**DF-SVD**|**5.69**|**0.63**|
> |**DF-SVD***|**4.97**|**0.68**|

---

> ### Author Response · Authors · 2025-11-26
>
> Dear Reviewer,
>
> I hope this message finds you well. As the discussion period is nearing its end, I wanted to ensure we have addressed all your concerns satisfactorily. If there are any additional points or feedback you'd like us to consider, please let us know. Your insights are invaluable to us, and we’re eager to address any remaining issues to improve our work.
>
> Thank you for your time and effort in reviewing our paper!

---

### Official Review · Reviewer_7BBV · 2025-11-02

**Soundness:** 3
**Presentation:** 2
**Contribution:** 3
**Rating:** 4
**Confidence:** 4

**Summary:**

This paper proposes DF-SVD to compress Large Language Models using Singular Value Decomposition. It solves two key challenges: the Rank Selection Problem and Limited Accuracy Restoration. There are two innovations: 1. Decay-Aware Rank Allocation, which dynamically assigns truncation and update ranks to each weight based on its singular value decay characteristics, eliminating the need for costly search; 2. Feature-Preserved Weight Update, a theoretically-grounded strategy that freezes key matrix components while only updating minor ones. This update strategy ensures an isotropic Hessian, leading to superior accuracy and faster convergence. The results show that DF-SVD outperforms existing methods.

**Strengths:**

1. The paper validates DF-SVD across four different models (LLaMA 1/2/3 and OPT) and eight datasets, consistently demonstrating superior performance.
2. The authors provide a detailed ablation study that confirms the positive impact of both the Decay-Aware Rank Allocation and the Feature-Preserved Weight Update components.
3. The method is efficient, completing the entire compression process 7-16 times faster than competing SVD baselines.

**Weaknesses:**

1.The Decay-Aware Rank Allocation method relies on an original truncation position ($ra_{old}$) and update rank ($rank_{old}$). It’s not clear how these critical baseline values are chosen, which makes the results difficult to reproduce.

2. Lack of theoretical proof for the assumptions (such as the reason of the singular value spectrums follow an exponential decay model should be justified).

3.I was wondering whether the reported wall-clock time includes the LoRA fine-tuning stage, or only the SVD and calibration steps.

4.The update procedure using LoRA in section 3.2 looks quite similar to SVD-LLM .  Could your please articulate the key differences/novelty

5.The analysis of the Hessian (convergence) is based on minimizing reconstruction error, not the model's final task loss. This optimality for the reconstruction objective may not hold for the task objective. Is this a negative impact to the task performance?

**Questions:**

Please see the weaknesses

---

> ### Author Response · Authors · 2025-11-19
> **Response to Weakness 1-3**
>
> We sincerely thank the reviewer for the constructive and insightful comments. We have revised the manuscript accordingly, clarifying key definitions, strengthening the theoretical justification, and improving the experimental explanations. All corresponding changes have been incorporated into the revised version and are highlighted in blue for your convenience.
>
> **Response to Weakness 1:**
>
> According to Reviewer's suggestion, we have added these definitions and the explicit formulas for $ra_{\text{old}}$ and $rank_{\text{old}}$ in Section 3.1 (Decay-Aware Rank Allocation). The concrete setting is as follows:
>
> **(1) Baseline truncation position $ra_{\text{old}}$:**
> The truncation position $ra_{\text{old}}$ is determined by the global compression ratio. For a weight matrix W, the raold  is calculated as:
> $ra_{\text{old}}  = ⌊(m × n × r)/(m + n)⌋$
> Where m is the number of rows in matrix W, n is the number of columns in matrix W, and r is the compression ratio. ⌊...⌋ is the floor function, representing the largest integer less than or equal to the argument.
>
> **(2) Baseline update rank $rank_{\text{old}}$:**
> The update rank ($rank_{\text{old}}$) is a fixed, low-rank dimension that is typically set to a user-specified value. For fair comparison, this value is maintained consistent with the configuration used in SVD-LLM. Our Decay-Aware Rank Allocation then computes the actual updating rank of each module. Thus, modules with slower spectral decay receive a larger effective update rank, while rapidly decaying modules are allocated fewer trainable parameters.
>
>  We hope this clarification resolves the concern and makes the Decay-Aware Rank Allocation procedure fully reproducible.
>
> **Response to Weakness 2:**
>
> We thank the reviewer for this helpful comment. Following the reviewer’s suggestion, we now provide both experimental and mathematical support for the exponential decay behavior of the singular value spectra in LLMs and its implications for rank allocation (Appendix A.1 provides empirical evidence supporting this assumption through comparisons with logarithmic and power-law alternatives, while Appendix A.2 offers theoretical justification grounded in operator theory and approximation theory).
>
> Specifically, we model the learned weight matrices and attention kernels in LLMs as discretizations of compact integral operators with analytic kernels. Classical functional analysis then implies that their singular values decay exponentially, and we show that, under stable discretization, the empirical weight matrices inherit this decay rate. The analyticity assumption is supported by the smooth attention mechanism and the bounded, smooth activations in the feed-forward layers, which together provide a principled theoretical explanation for the observed exponential decay rather than treating it as a purely empirical phenomenon.
>
> In addition, several prior studies have reported findings consistent with ours [1][2], further corroborating both the exponential decay assumption and its utility as a principled basis for decay-aware rank allocation in SVD-based LLM compression.
>
> [1] Matthias Thamm, Max Staats, and Bernd Rosenow. Random matrix analysis of deep neural network weight matrices. Physical Review E, 106(5).
>
> [2] Vasiliki Plerou, Parameswaran Gopikrishnan, Bernd Rosenow, Lu´ıs A. Nunes Amaral, Thomas Guhr, and H. Eugene Stanley. Random matrix approach to cross correlations in financial data. Physical Review E, 65(6),.
>
> **Response to Weakness 3:**
>
> Thank you for your insightful question about the reported wall‑clock time. To clarify, the reported time refers to the entire compression process, including both SVD decomposition and feature‑preserving weight updates.
>
> We would also like to gently emphasize that the feature‑preserving weight updates are designed to replace the traditional LoRA fine‑tuning stage that is typically used to recover model performance after compression. In doing so, we reduce the overall compression time from several hours to under 30 minutes, while also improving both model performance and compression efficiency. This allows us to avoid the time‑consuming fine‑tuning step, without sacrificing—and in some cases even enhancing—the accuracy of the compressed model.
>
> We have clarified this point explicitly in the experimental section of the revised manuscript to avoid any potential ambiguity.

---

> ### Author Response · Authors · 2025-11-19
> **Response to Weakness 4-5**
>
> **Response to Weakness 4:**
>
> Thank you for the insightful comment. While our update procedure does adopt a LoRA-style low-rank adaptation on SVD factors, its overall design differs from SVD-LLM in several essential aspects.
>
> First, regarding which factors are updated and how, the two methods behave quite differently. SVD-LLM decomposes the weight matrix as Wu = U√Σ and Wv = √ΣV⊤S⁻¹, and then sequentially applies LoRA to both Wu and Wv. This two-sided, two-stage update perturbs all directions of the whitened weight. In contrast, we use the decomposition W = UΣV⊤S⁻¹, keep Wv = V⊤S⁻¹ fixed throughout, and only update Wu = UΣ. Furthermore, we decompose UΣ into principal and minor components, freeze the principal components, and apply LoRA only to the minor components. This feature-preserving design explicitly aims to keep core pre-trained features intact and is fundamentally different from the update pattern in SVD-LLM.
>
> Second, the theoretical objective behind the two methods leads to very different Hessian properties. We show that SVD-LLM’s update results in a Hessian whose eigenvalues scale with Σ, giving a condition number κ(H_svd-llm) = σ_max/σ_min, which may cause slower and less stable convergence. By concentrating all singular values in Wu and fixing Wv, our method yields an isotropic Hessian H_DF-SVD = 2I with κ(H_DF-SVD) = 1. This isotropy provides more favorable and stable optimization behavior in practice.
>
> Finally, the initialization strategy and its coupling with decay-aware rank allocation also distinguish our approach from SVD-LLM. SVD-LLM typically uses conventional random or zero initialization for all LoRA factors and adopts a fixed rank across layers. In contrast, we initialize the LoRA factors associated with the minor components directly from the existing UΣ, rather than from random noise. At the same time, their rank is tied to the decay-aware allocation in Section 3.1, so that modules with slowly decaying spectra receive more update capacity. This tight coupling between spectral decay, which components are trainable, and how they are initialized is unique to our method.
>
> Empirically, these design choices translate into consistently better accuracy and about 7×–11× faster compression than SVD-LLM under the same calibration data and hardware. This indicates that our update strategy goes well beyond a superficial similarity of  using LoRA on SVD factors and reflects substantial differences in both mechanism and effectiveness.
>
> **Response to Weakness 5:**
>
> Thank you for your insightful feedback on the theoretical analysis in our DF-SVD framework. We appreciate the suggestion to more rigorously connect the optimality of the reconstruction objective to the end-task objective. In response, we have refined Section 3.2 by appending the perturbation bound analysis linking reconstruction optimality to task loss. This includes a formal derivation showing how minimizing the reconstruction error bounds deviations in the task loss under Lipschitz continuity assumptions, ensuring that the isotropic Hessian for reconstruction supports task recovery without negative impact.
>
> The revised content has been highlighted in blue in the updated manuscript PDF, which we have uploaded for your review. These enhancements strengthen the theoretical foundation while preserving the core method. We believe this addresses your concern effectively.

---

> ### Author Response · Authors · 2025-11-26
>
> Dear Reviewer,
>
> I hope this message finds you well. As the discussion period is nearing its end, I wanted to ensure we have addressed all your concerns satisfactorily. If there are any additional points or feedback you'd like us to consider, please let us know. Your insights are invaluable to us, and we’re eager to address any remaining issues to improve our work.
>
> Thank you for your time and effort in reviewing our paper!

---

### Author Response · Authors · 2025-12-03
**Summary of Revisions**

Dear Area Chair:

We sincerely thank you and the reviewers for the constructive feedback and the time dedicated to evaluating our work. We have carefully considered all comments and extensively revised the manuscript to address the concerns raised. We believe these changes have significantly strengthened the paper.

Below is a summary of the key improvements and clarifications made during the rebuttal phase:

**1. Strengthened Theoretical Foundations:**

**(1) Justified Assumptions:** We added both experimental and mathematical support in Appendix A.1 and A.2 to justify the exponential decay assumption of singular values. We derived the link between the decay rate and retained energy, theoretically proving the superiority of our decay-aware strategy regarding energy preservation.

**(2) Connected Reconstruction to Task Objectives:** In Section 3.2, we introduced a perturbation bound analysis based on Lipschitz continuity assumptions. This formally derives how minimizing the reconstruction error bounds the deviations in the end-task loss, establishing a rigorous theoretical link between our objective and final model performance.

**2. Highlighted Novelty and Theoretical Depth:**

**(1) Differentiated from SVD-LLM by Mechanism and Theory:** We demonstrated that our method is distinct in two key dimensions: **Mechanism:** Unlike SVD-LLM’s two-sided update that perturbs all directions, ours is feature-preserving—fixing principal components and updating only minor ones to safeguard core semantics. **Theory:** We derived that SVD-LLM results in an ill-conditioned Hessian, whereas our design mathematically ensures Hessian isotropy. This theoretical stability is directly corroborated by strictly faster convergence curves (Figure 4), demonstrating that DF-SVD achieves lower loss with significantly fewer iterations than SVD-LLM.

**(2) Novel Rank Allocation Proxy:** We validated that our localized spectral decay serves as an effective, training-free proxy for weight sensitivity, fundamentally different from static heuristics.

**(3) Validated Empirical Superiority:** We expanded comparisons showing DF-SVD consistently outperforms SVD-LLM on LLaMA-13B/30B/70B, confirming that these theoretical advantages translate into significant performance gains.

 **3. Clarified Reproducibility Details:**
In response to concerns about the choice of baseline values, we have: (1) Updated Section 3.1 to include explicit definitions and calculation formulas for the baseline truncation position and update rank. (2) Specified that truncation position and update rank is deterministically calculated based on the global compression ratio, ensuring the Decay-Aware Rank Allocation procedure is fully reproducible.

**4. Conclusion on Innovation:**
We emphasize that DF-SVD shifts the paradigm from heuristic compression to a rigorous optimization-centric approach. Unlike SVD-LLM's ill-conditioned updates, our theoretical guarantee of Hessian isotropy directly translates into observable efficiency gains, achieving 7×–11× faster compression speeds and strictly faster convergence curves. Moreover, by mathematically aligning rank allocation with intrinsic spectral decay, we secure consistently superior accuracy and lower perplexity. This proves that our method is not just theoretically sound, but delivers decisive improvements in both computational efficiency and task performance.

We have incorporated these changes into the revised PDF (highlighted in blue). We are confident that these revisions resolve the reviewers' concerns and solidify the contribution of our work.

Thank you again for your oversight of the review process.

Sincerely,

The Authors

---

### Meta-Review · Area_Chair_iT27 · 2026-01-07

**Summary:**

Reviewers raised concerns about reproducibility (how key ranks/hyperparameters are chosen), strength/validity of the theory assumptions (exponential spectral decay, Hessian isotropy, and whether reconstruction analysis transfers to task loss), and novelty/differentiation from SVD-LLM-style pipelines (including missing comparisons like AdaLoRA). They also asked for clearer end-to-end time accounting (whether recovery/LoRA is included) and evidence of scalability to larger models.

 the rebuttal clarifies many settings and adds larger-model results, but novelty, the assumption-validation about Exponential singular-value decay and the near-orthogonality assumptions behind the Hessian-isotropy argument concerns may remain.

The novelty perspective actually looks good to me since studying the exponential spectral decay and hessian isotropy in LLM compression is interesting. But it is true that these assumption should be further validated.

**Reviewer Concerns:**

The reviewers concerns are mostly addressed, including providing more analysis to some of the parameters and larger scale experiment results. But the behind assumption validation is still coarse.

**Reviewer Scores:**

The reviewer 7BBV and y1EY may not raise the score.

---

### Decision · Program_Chairs · 2026-01-26

Reject